# Development of a New Chemiluminescent Enzyme Immunoassay Using a Two-Step Sandwich Method for Measuring Aldosterone Concentrations

**DOI:** 10.3390/diagnostics11030433

**Published:** 2021-03-04

**Authors:** Yoshinori Ozeki, Yukie Tanimura, Satoshi Nagai, Takuya Nomura, Mizuki Kinoshita, Kanako Shibuta, Naoki Matsuda, Shotaro Miyamoto, Yuichi Yoshida, Mitsuhiro Okamoto, Koro Gotoh, Takayuki Masaki, Kengo Kambara, Hirotaka Shibata

**Affiliations:** 1Department of Endocrinology, Metabolism, Rheumatology and Nephrology, Faculty of Medicine, Oita University, Yufu City 879-5593, Oita, Japan; ozeki23@oita-u.ac.jp (Y.O.); tanimuray@oita-u.ac.jp (Y.T.); n-satoshi@oita-u.ac.jp (S.N.); ta-nomura@oita-u.ac.jp (T.N.); mogawa@oita-u.ac.jp (M.K.); kanako-o@oita-u.ac.jp (K.S.); matsuda20656@oita-u.ac.jp (N.M.); shoutarou1029@oita-u.ac.jp (S.M.); y-yoshida@oita-u.ac.jp (Y.Y.); mokamoto@oita-u.ac.jp (M.O.); gotokoro@oita-u.ac.jp (K.G.); masaki@oita-u.ac.jp (T.M.); 2FUJIFILM Wako Pure Chemical Corporation, Amagasaki City 661-0963, Hyogo, Japan; kengo.kambara@fujifilm.com

**Keywords:** aldosterone concentration, CLEIA, two-step sandwich method, primary aldosteronism

## Abstract

In the present study, we developed a new chemiluminescent enzyme immunoassay (CLEIA) using a two-step sandwich method to measure aldosterone concentrations. We investigated serum and plasma aldosterone concentrations in 75 blood samples from 27 patients using a radioimmunoassay (RIA) and the CLEIA (with current and newly improved reagents) as well as liquid chromatography-tandem mass spectrometry (LC-MS/MS). Based on the results of the Passing–Bablok regression analysis, the aldosterone levels measured using CLEIA with the new reagents and those measured by LC-MS/MS were found to be significantly correlated (slope, 0.984; intercept, 0.2). However, aldosterone levels varied depending on the measurement method (i.e., CLEIA with the new reagent, CLEIA with the current reagent, and RIA). Aldosterone levels were lower with the improved CLEIA method than with RIA and CLEIA using the current reagent. Therefore, the cutoff values of the screening test as well as those of the confirmatory test for primary aldosteronism (PA) should be adjusted to follow current clinical practice guidelines for PA. The formula that can be used to obtain the aldosterone level (pg/mL) when using CLEIA with the new reagent is 0.765 × RIA (pg/mL) − 33.7. This formula will enable PA cutoff values to be set for provisional screening and confirmatory tests.

## 1. Introduction

Primary aldosteronism (PA) is the most frequent cause of secondary hypertension, and 5–10% of patients with hypertension have PA [1,2,3]. PA causes hypertension and hypokalemia due to excessive aldosterone secretion by the adrenal glands [4,5].

Patients with PA have an increased risk of cardiovascular events, such as cerebral infarction, myocardial infarction, and chronic kidney disease, compared to those with essential hypertension, and early diagnosis makes PA a treatable disease [6,7,8,9].

PA is diagnosed via a screening test, followed by a confirmatory or exclusion test and subtype test using computed tomography and adrenal vein sampling (AVS). The aldosterone to renin ratio (ARR) is used as a screening test for PA [2,10,11]. In Japan, the screening cutoff values for PA diagnosis are a plasma aldosterone concentration (PAC) > 120 pg/mL and a PAC to plasma renin activity ratio of >200 or a PAC to active renin concentration ratio (ARC) of >40.

Subsequently, we performed an oral sodium loading test (OSLT), captopril challenge test (CCT), saline infusion test (SIT), and furosemide upright posture test to confirm or exclude the diagnosis of PA [2,6,8,11,12]. When at least one confirmatory test was positive, we diagnosed the patient with PA. Lastly, when a patient wants to undergo curative adrenal surgery, it is crucial to conduct subtype testing using AVS. In clinical practice, “accurate measurement of PAC is crucial because entire diagnostic tests critically depend on aldosterone concentrations”.

The PAC and ARC are conventionally measured by radioimmunoassay (RIA) [13,14]. However, the currently available chemiluminescent enzyme immunoassay (CLEIA) makes it possible to simultaneously measure PAC and ARC more rapidly [15]. In addition, the CLEIA enables accurate measurement of aldosterone levels in plasma and serum samples. Although a relatively strong correlation has been observed for aldosterone concentrations between RIA and CLEIA, there may be some discrepancy in aldosterone concentrations between the results of CLEIA and liquid chromatography-tandem mass spectrometry (LC-MS/MS) tests [16]. Therefore, we developed a new CLEIA kit to measure aldosterone concentrations using a two-step sandwich method. Aldosterone concentrations derived using RIA, the current CLEIA method, the new CLEIA method, and LC-MS/MS from patients with and without PA were compared. The results showed that the aldosterone concentrations measured with the new CLEIA kit were highly correlated with those measured with LC-MS/MS; moreover, the concentrations were lower than those obtained using the current CLEIA kit and RIA.

## 2. Materials and Methods

### 2.1. Patients and Study Design

We collected 75 blood samples from 27 patients (11 males and 16 females) hospitalized in Oita University Hospital between July and August 2020. We diagnosed 15 patients with PA based on a positive screening test and at least one positive confirmatory test for PA (PA group). Twelve obese diabetic patients were not diagnosed with PA based on negative PA screening results (non-PA group). The screening test was considered positive when the PAC/ARC (pg/mL) ratio was >40 and the PAC was >120 pg/mL. The SIT involved intravenous infusion of 2 L of saline over 4 h, with the patient in a supine position. The test was considered positive when the PAC was >60 pg/mL after saline loading. The CCT was performed by administering 50 mg of captopril, with blood drawn after 90 min. When the PAC/ARC ratio was >40 at 90 min after administration, the test was considered positive. The OSLT results were determined through the analysis of 24-h urine samples after consumption of a high-salt diet. Tests were considered positive when the 24-h urinary aldosterone and Na excretion levels were >8 μg/day and >170 mEq/day, respectively. A urine Na concentration of <170 mEq/day was considered to reflect insufficient NaCl loading; those cases were excluded from the analysis.

In the present study, 17 of 20 patients with hypertension were taking antihypertensive medications (calcium channel blockers, 16 patients; angiotensin II receptor blockers, 5 patients; or a mineralocorticoid receptor antagonist, 1 patient).

Blood samples were collected in tubes containing EDTA-2Na, including 24 samples taken in the early morning, 8 before the SIT, 8 after the SIT, 2 before the CCT, and 2 after the CCT, as well as 31 samples collected during the AVS. In 36 cases, serum and plasma samples were collected simultaneously, while only serum was collected in 39 cases (total serum samples, n = 75 samples; total plasma samples, n = 36).

We measured aldosterone concentrations in samples using RIA and CLEIA (with the current reagent and the improved reagent) as well as LC-MS/MS, which is the gold standard measurement method according to The Japan Endocrine Society [17,18].

This study design followed the Declaration of Helsinki and was approved by the Ethical Committee of Oita University. All subjects gave informed consent to participate in the study.

### 2.2. Radioimmunoassay

The SPAC-S aldosterone kit (Fujirebio Co., Ltd., Tokyo, Japan) is based on a competitive RIA method. The aldosterone in the sample competes with iodine 125 (tracer)-labeled aldosterone for the antibody coated on the tubes. After aspiration, the level of radioactivity in the tubes is measured with a gamma counter. The degree of binding is inversely proportional to the aldosterone concentration in the sample. The detection range used in this study was 25–1600 pg/mL.

### 2.3. CLEIA Method Using the Current Reagent

We use the Accuraseed aldosterone kit (FUJIFILM Wako Pure Chemical Corporation, Osaka, Japan) in conjunction with the CLEIA method for routine aldosterone testing.

A 35 μL aliquot of the sample and 30 μL of an immune reaction buffer were added to 25 μg of antibody-bound particles and reacted at 37 ℃ for about 3 min. Next, 25 μL of enzyme-labeled antigen was added, and the mixture was reacted at 37 ℃ for about 3 min. Then, the bound and free fractions were separated. Finally, 50 μL of substrate solution and 100 μL of hydrogen peroxide solution were added, and the amount of light emitted per unit time was measured. Using the standard solution as a sample, the amount of luminescence per unit time was measured using the same steps to create a calibration curve. The amount of luminescence in the sample was applied to the calibration curve to calculate the aldosterone concentration. The detection range was 50–1600 pg/mL.

### 2.4. CLEIA Method Using the Improved Reagent

The improved reagent, Accuraseed aldosterone S kit (FUJIFILM Wako Pure Chemical Corporation), was used with the CLEIA method. The antibody bound particle–aldosterone–enzyme-labeled antibody immune complex was formed when aldosterone in the sample reacted with the anti-aldosterone monoclonal antibody (mouse)-bound particles and then with a peroxidase-labeled anti-aldosterone immune complex monoclonal antibody (rat).

As the amount of enzyme bound to the antibody-bound particles was proportional to the aldosterone concentration, the aldosterone concentration in the sample was determined by measuring enzyme activity using a chemiluminescent reagent (luminol, hydrogen peroxide).

The low level of reproducibility was improved by using an antibody clone with high antigen affinity and little cross-reactivity to the magnetic particles, and this was combined with the newly developed clone. A 25 μL aliquot of the sample and 50 μL of immune reaction buffer were added to 25 μg of antibody-bound particles, and the mixture was reacted at 37 ℃ for about 3 min. Then, the bound and free fractions were separated. Next, 50 μL of enzyme-labeled antibody was added, and the mixture was reacted at 37 ℃ for about 3 min, followed by separation of the bound and free fractions. Finally, 100 μL of the substrate solution and 100 μL of the hydrogen peroxide solution were added, and the amount of light emitted per unit time was measured. Using the standard solution as a sample, the amount of luminescence per unit time was measured using the same steps to create the calibration curve. The sample luminescence was applied to the calibration curve to calculate the aldosterone concentration. We drew a normal positive standard curve using this two-step sandwich assay, and precision and sensitivity were both found to be improved. Furthermore, the reaction was unaffected by the sample ingredients, by reducing the sample volume from 35 to 25 μL, or by separating the bound/free fractions twice. A schematic representation of the measurements is shown in Figure 1.

### 2.5. LC-MS/MS

As a control, we compared the improved reagent results with those obtained with LC-MS/MS. We consigned the LC-MS/MS measurements to ASKA Pharmaceutical Co., Ltd. (Tokyo, Japan). Their LC-MS/MS system is traceable to NMIJ CRM 6402. The concentration of the certified value of CRM 6402 was determined by ID-LC/MS/MS, which is the reference measurement procedure for isotope dilution-mass spectrometry [17,18].

### 2.6. Statistical Analysis

The data are presented as the mean ± standard deviation, and differences between groups were assessed with Student’s *t*-test. Statistical significance was considered at *p* < 0.05. The correlations were analyzed by the Passing–Bablok regression analysis and the Bland–Altman analysis using XLSTAT (Addinsoft, New York, NY, USA).

## 3. Results

### 3.1. Basal Clinical Characteristics of the Patients

The clinical features of the subjects are shown in Table 1. Body mass index values were significantly higher in the non-PA patients than the PA patients. The systolic blood pressure and diastolic blood pressure values were significantly higher in the PA patients than the non-PA patients. No significant differences were observed in the PAC or ARC between groups, but the ARR was significantly higher in PA patients than in non-PA patients.

### 3.2. Validation of Aldosterone Concentrations Using the Improved Reagent

Detailed verification data regarding the improved reagent are shown in the Appendix A. The lower-range measurement accuracy improved compared with that obtained with the current reagent (Appendix A), and the upper limit of measurement expanded to 3200 pg/mL [15]. As a result, the detection range was 3–3200 pg/mL. It was necessary to measure the aldosterone concentration in the AVS samples after dilution, because many of the adrenal venous samples had very high aldosterone levels. Expansion of the upper limit of measurement could make it possible to reduce the number of dilutions.

### 3.3. Serum and Plasma Aldosterone Concentrations Using the Improved Reagent

The analysis was performed on 36 plasma and serum samples that were collected at the same time. The results of the Passing–Bablok regression analysis of the serum and plasma aldosterone concentrations measured using the improved reagent are shown in Figure 2A. The slope was 1.026 and the intercept was 0.2, and 95% confidence intervals were also calculated.

### 3.4. Correlation of Aldosterone Concentrations between the Improved Reagent and RIA

The analysis was performed using 75 serum samples. The Passing–Bablok regression analysis between the aldosterone concentrations obtained using the improved reagent (new CLEIA aldosterone assay) and the RIA is shown in Figure 2B,C. The slope was 0.857 the intercept was −44.9, and 95% confidence intervals were calculated. The aldosterone level measured with the new CLEIA assay was lower than that measured with RIA. The formula to obtain the aldosterone level using the new CLEIA aldosterone assay, based on 56 samples that had an aldosterone level < 1600 pg/mL according to the RIA, is 0.765 × RIA − 33.7 (Figure 2C).

### 3.5. Correlation of Aldosterone Concentrations Obtained with the Improved and Current Reagents

This analysis was performed using 75 serum samples. The Passing–Bablok regression analysis of the aldosterone concentrations obtained using the new CLEIA aldosterone assay with the improved reagent and the CLEIA assay with the current aldosterone reagent are shown in Figure 2D,E. The slope was 0.776, the intercept was −27.5, and 95% confidence intervals were calculated. Similar to the RIA, the aldosterone levels obtained using the new CLEIA assay were lower than those obtained with the current CLEIA assay.

### 3.6. Correlation of Aldosterone Concentrations between the Improved Reagent and LC-MS/MS

This analysis was also performed using 75 serum samples. The Passing–Bablok regression analysis between the aldosterone concentrations obtained with the new CLEIA assay and LC-MS/MS are shown in Figure 2F, G. The slope was 0.984, the intercept was 0.2, and 95% confidence intervals were calculated. The Bland–Altman analysis showed that the mean difference in the aldosterone concentration between the two assays was −2.7 pg/mL, with a 95% confidence interval of −8.2 to 2.8 pg/mL in a concentration range of up to 1600 pg/mL (Figure 3).

## 4. Discussion

PA is a highly prevalent and treatable form of hypertension; therefore, prompt screening and diagnosis are required [1,2,3]. The supply of PAC assay kits using the RIA method will be terminated in March 2021 in Japan; therefore, non-RIA measurement methods will become important. Further improvements in aldosterone measurement accuracy are expected, as a new antibody has recently been developed to measure aldosterone concentrations using a CLEIA two-step sandwich method. The present study demonstrated that the CLEIA two-step sandwich method could be used in clinical practice in conjunction with a new reagent.

We collected 75 blood samples from 27 patients (PA group, 15 patients; non-PA group 12 patients). We did not detect any significant differences in the PAC or ARC between the two groups, because more patients were given angiotensin II receptor blockers in the non-PA patient group than in the PA group, and more obese patients were included in the non-PA than the PA group. Second, we could not prove that the patients in the latter group did not have PA, because we did not perform confirmatory testing.

The aldosterone levels measured using the new CLEIA assay and LC-MS/MS were significantly correlated, and the lower-range measurement accuracy was improved. Therefore, the new CLEIA aldosterone assay could be useful in clinical practice, where quick and accurate measurement of aldosterone levels is needed.

A difference in the results was observed for the new CLEIA aldosterone assay and LC-MS/MS compared with RIA and the current CLEIA aldosterone assay. The measured aldosterone levels were lower with the new CLEIA aldosterone assay and LC-MS/MS than with the RIA and current CLEIA aldosterone assays. Therefore, a new screening cutoff value for PA and specific criteria for the confirmatory test may be required [19]. Based on our new formula used for the new CLEIA aldosterone assay (0.765 × RIA − 33.7), provisional screening cut-off values as well as the confirmatory test for PA need to be adjusted. Four of the eight blood samples collected after the SIT tested positive (PAC ≥ 60 pg/mL) with the CLEIA method using the current reagent, but only one of these eight samples tested positive with the improved reagent. The provisional cutoff value for the SIT based on our new formula was 12.2 pg/mL. Using this cutoff value and the improved reagent, seven of the eight samples tested positive. In addition, six of the eight samples tested positive with LC-MS/MS. The screening test cutoff value for the RIA was PAC > 120 pg/mL, but our new formula gave a value of 58.1 pg/mL. However, as this result was obtained based on a small number of cases, further verification is required.

After diagnosing PA with a confirmatory test, AVS is the gold standard test for diagnosing the PA subtype [20,21,22,23,24]. According to The Japan Endocrine Society, the most commonly used AVS cutoff values after adrenocorticotropic hormone stimulation are a lateralized ratio (LR) of >4 and a contralateral ratio (CR) of <1. The LR is the ratio of aldosterone to cortisol in the dominant adrenal vein compared with that in the nondominant adrenal vein. The CR is the ratio of aldosterone to cortisol in the nondominant adrenal vein compared with that in the inferior vena cava. Aldosterone was present at low and high concentrations in the AVS samples. It was necessary to measure aldosterone levels after dilution because many of the adrenal venous samples had very high aldosterone levels (>3200 pg/mL). First, we analyzed the undiluted adrenal venous samples. If the aldosterone level was beyond the upper limit of the assay, we analyzed the sample again after conducting a 15-fold dilution. In this study, there were very limited cases in which blood samples needed to be diluted to determine the aldosterone concentration using the new CLEIA method, because the upper detection limit was 3200 pg/mL, which is equivalent to a value of about 4300 pg/mL with RIA after conversion. This would further reduce the re-examination rate. The expanded measurement range with the new method enables earlier confirmation of the aldosterone concentration than the current method does and requires only 10 min to obtain a result. Thus, the new method is suitable for the analysis of adrenal venous samples.

Further study is needed to determine the optimal screening cut-off values as well as the PA confirmatory test values using the new CLEIA aldosterone assay.

The present study had several limitations. First, the number of cases was small. Second, many of the participants were taking drugs that affect aldosterone levels. Third, urine samples should also be evaluated. These limitations should be addressed in a future study.

## Figures and Tables

**Figure 1 diagnostics-11-00433-f001:**
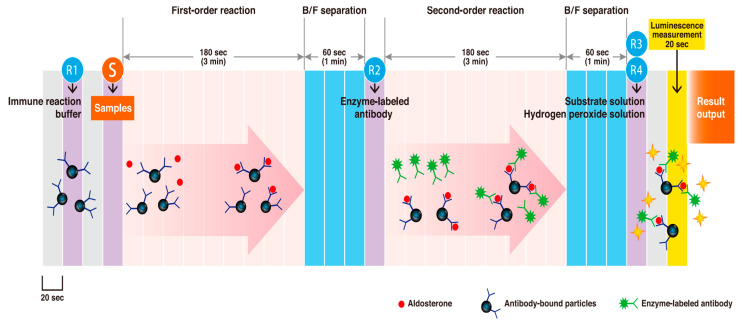
Schematic representation of aldosterone concentrations measurement using a two-step sandwich method.

**Figure 2 diagnostics-11-00433-f002:**
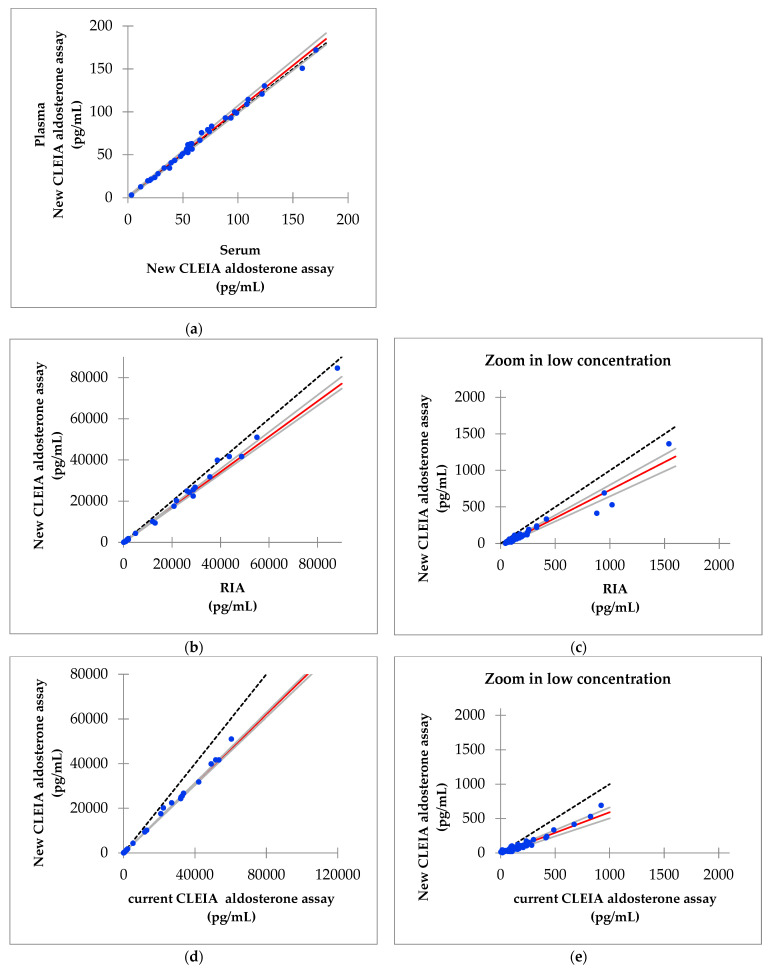
Results of the Passing−Bablok regression. (**a**) Serum versus plasma aldosterone concentrations measured using the improved reagent; (**b**) aldosterone concentrations measured using the improved reagent (new chemiluminescent enzyme immunoassay (CLEIA) aldosterone assay) versus the radioimmunoassay (RIA); (**c**) aldosterone concentrations measured using the improved reagent versus the RIA in a concentration range of up to 1600 pg/mL; (**d**) aldosterone concentrations measured using the new CLEIA aldosterone assay versus the current CLEIA aldosterone assay; (**e**) aldosterone concentrations measured using the new versus current CLEIA aldosterone assay in a concentration range of up to 1600 pg/mL; (**f**) aldosterone concentrations measured using the new CLEIA assay versus LC-MS/MS; (**g**) aldosterone concentrations measured using the new CLEIA assay versus LC-MS/MS in a concentration range of up to 1600 pg/mL.

**Figure 3 diagnostics-11-00433-f003:**
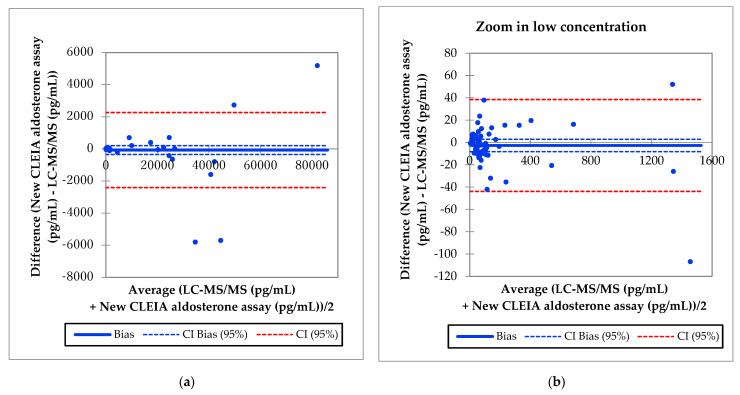
Bland-Altman results. (**a**) Aldosterone concentrations measured using the new CLEIA assay versus LC-MS/MS; (**b**) aldosterone concentrations measured using the new CLEIA assay versus LC-MS/MS in a concentration range of up to 1600 pg/mL.

**Table 1 diagnostics-11-00433-t001:** Basal clinical characteristics of participants in the study.

	PA	Non-PA	*p*
Age (years)	51.5 ± 13.5	47.0 ± 16.3	0.45
Male/Female	6/9	5/7	
BMI (kg/m^2^)	23.5 ± 2.9	30.8 ± 8.1	0.01
Systolic blood pressure (mmHg)	129.2 ± 15.2	113.1 ± 11.2	0.004
Diastolic blood pressure (mmHg)	84.3 ± 11.3	72.3 ± 15.1	0.03
HR (bpm)	76.6 ± 13.5	78.9 ± 10.3	0.61
BUN (mg/dL)	11.2 ± 3.3	15.2 ± 5.9	0.05
Creatinine (mg/dL)	0.68 ± 0.11	0.89 ± 0.34	0.06
Na (mmol/L)	139.9 ± 1.7	139.3 ± 3.0	0.50
K (mmol/L)	3.8 ± 0.3	4.0 ± 0.4	0.23
Cl (mmol/L)	106.0 ± 2.0	104.4 ± 2.5	0.09
eGFR (mL/min/1.73 m^2^)	82.7 ± 12.2	70.9 ± 24.0	0.14
Plasma aldosterone concentration (pg/mL)	158.9 ± 57.0	170.0 ± 84.6	0.70
Active renin concentration (pg/mL)	7.4 ± 19.2	28.0 ± 31.2	0.06
ARR	103.2 ± 76.3	12.5 ± 9.5	<0.001
Antihypertensive drugs			
Calcium channel blocker	10/15 (67%)	6/12 (50%)	
Angiotensin II Receptor Blocker	0/15 (0%)	5/12 (42%)	
Mineralocorticoid receptor antagonist	1/15 (7%)	0/12 (0%)	

Date are shown as average (mean ± SD); BMI: Body mass index, PA: Primary aldosteronism, ARR: Aldosterone-to-Renin Ratio.

## Data Availability

The date presented in this study are available on request from the corresponding author.

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
