# Peer review of "Development of a New Chemiluminescent Enzyme Immunoassay Using a Two-Step Sandwich Method for Measuring Aldosterone Concentrations"

_diagnostics, 2021, doi:10.3390/diagnostics11030433_

Round 1
Reviewer 1 Report
Yoshinori Ozeki et al. developed a new chemiluminescent immunoassay for the measurement of aldosterone in plasma and serum samples. The results obtained with the new assay were compared with the results of different available assays (radioimmunoassay, LC-MS/MS and a currently used chemiluminescent assay). It has been convincingly demonstrated that the new method has better reproducibility and a wider measurement range compared to the currently used chemiluminescent method. In addition, the results of the new method showed better agreement with the results obtained using the LC-MS / MS method, which is accepted as a gold standard.
There are minor issues that require to be addressed before proceeding with the publication.
1/ Page 2, line 83: What type of anticoagulant was used to collect the plasma samples?
2/ Figure 1. shows the schematic representation of aldosterone measurement instead of “Schematic diagram of aldosterone concentrations measured..”
3/ In Table S1, the LOQ is set to be 3 pg / ml. In contrary, on page 5 (line 165) the lower limit of the detection range is set to be 5 pg/ml. What is the reason for this discrepancy?
4/ In the Supplementary Material, in chapter “Limit of blank, limit of detection and limit of quantitation for the improved assay” the LoD value (4 pg/ml) is not correspond to the value in Table S1. (1.3 pg/ml).
5/ The description of the recovery test is confusing.
6/ How much cross-reactive substances were added to the low-concentration pooled plasma sample during the investigation of cross-reactivity?
7/ A language and punctuation check is highly recommended.
Author Response
Response to Reviewer 1 Comments
Thank you very much for your suggestions.
We present point-by-point responses to your questions and comments.
Point 1: Page 2, line 83: What type of anticoagulant was used to collect the plasma samples?
Response 1: We added the following to Page2, line 83.
Blood samples were collected in tubes containing EDTA-2Na・・
Point 2: Figure 1. shows the schematic representation of aldosterone concentrations measurement instead of “Schematic diagram of aldosterone concentrations measured..”
Response 2: We revised this to “schematic representation of aldosterone concentrations measurement using a two-step sandwich method”.
Point 3: In Table S1, the LOQ is set to be 3 pg / ml. In contrary, on page 5 (line 165) the lower limit of the detection range is set to be 5 pg/ml. What is the reason for this discrepancy?
Response 3: This was a clerical error. We used the LoQ for the lower limit of measurement. Therefore, the lower limit of the detection range was 3 pg/mL. We have corrected this in the text.
Point 4: In the Supplementary Material, in chapter “Limit of blank, limit of detection and limit of quantitation for the improved assay” the LoD value (4 pg/ml) is not correspond to the value in Table S1. (1.3 pg/ml).
Response 4: This was a clerical error. We corrected the text to 1.3 pg/mL.
Point 5: The description of the recovery test is confusing.
Response 5: We fixed Supplementary Material Page 2 as follows.
The test was performed using base samples (Sample Series A) and aldosterone solutions (Sample series B) at a volume ratio of 9:1. The control samples were prepared similarly by adding specimen diluent to the base samples. The recovery rates were 92.5–107.8% (Table S7).
Point 6: How much cross-reactive substances were added to the low-concentration pooled plasma sample during the investigation of cross-reactivity?
Response 6: The concentration used was 100 μg/mL. This information has been added to Table S9.
Point 7: A language and punctuation check is highly recommended.
Response 7: Our manuscript has undergone language editing and proofreading by MDPI.
Having made these changes to our final manuscript, we would like to resubmit it.
Thank you once again for your consideration of our paper.
Yoshinori Ozeki

Reviewer 2 Report
The manuscript reports the application of a modified commercial method for the determination of aldosterone for the diagnosis of primary aldosteronism.
Since the work presents the comparison of different measurement methods, this should be done with an adequate number of samples with directly measurable concentrations (without further dilutions) and homogeneously distributed in the concentration range of the calibration curve.
Since a different antibody is used than the one validated in the original method, data on the maximum concentration level should be reported for which a "hook effect" can be excluded.
The incredible specificity, sensitivity and repeatability characteristics of the improved reagent should be compared to those of the original reagent. In particular, given that it is stated that the accuracy characteristics have improved (line 224), what reported in the supplementary data regarding cross reactivity (for example especially with regard to cortisol) should be demonstrated in more detail.
The formula indicated in the summary for the conversion of results is not explained in the text, furthermore it is not proven whether there is a constant or proportional bias between the methods to the level of concentration.
For the method in mass spectrometry, used as a reference, a greater detail of the type of instrumentation and analytical procedures used should be indicated: in particular the material used for the calibration of the system.
References: items 16 and 18 are the same.
The following reference should be also cited: K. Teruyama et al. "Novel chemiluminescent immunoassay to measure plasma aldosterone and plasma active renin concentrations for the diagnosis of primary aldosteronism" Journal of Human Hypertension https://doi.org/10.1038/s41371-020-00465-5
Author Response
Response to Reviewer 2 Comments
Thank you very much for your suggestions.
We present point-by-point responses to your questions and comments.
Point 1: Since the work presents the comparison of different measurement methods, this should be done with an adequate number of samples with directly measurable concentrations (without further dilutions) and homogeneously distributed in the concentration range of the calibration curve. 

Response 1: The directly measurable concentration (without further dilutions) of the new CLEIA reagent was 3–3200 pg/mL. Certainly, there were a few samples with higher concentrations. However, equivalent correlations were obtained at concentrations of up to 3200 pg/mL and above (Fig.2F). Furthermore, the dilution linearity of plasma samples was confirmed at up to 3569.9 pg/mL. Based on these results, we think that this concentration was appropriate.
Point 2: Since a different antibody is used than the one validated in the original method, data on the maximum concentration level should be reported for which a "hook effect" can be excluded.
Response 2: We added data on the hook effect (Fig S4). The hook effect is not observed at concentrations of up to 1,000,000 pg/mL. None of the 31 samples in the AVS exceeded concentrations of 1,000,000 pg/mL.
Point 3: The incredible specificity, sensitivity and repeatability characteristics of the improved reagent should be compared to those of the original reagent. In particular, given that it is stated that the accuracy characteristics have improved (line 224), what reported in the supplementary data regarding cross reactivity (for example especially with regard to cortisol) should be demonstrated in more detail.
Response 3: We added data showing the correlation of aldosterone concentrations between the original reagents and the LC-MS/MS and RIA (Fig S5). These were the data obtained at this time. The specificity, sensitivity, and repeatability characteristics of the original reagents were explained by citing Reference 15. We also added the tested concentrations of cross-reactive substances and the calculation method used to the text. These data show that the new reagent has an improved performance compared with the original reagent.
Point 4: The formula indicated in the summary for the conversion of results is not explained in the text, furthermore it is not proven whether there is a constant or proportional bias between the methods to the level of concentration.
Response 4: For the sake of clarity, the “SPAC-S” notation will be changed to “RIA”. This has been added the formula on Page9, line 280.
Point 5: For the method in mass spectrometry, used as a reference, a greater detail of the type of instrumentation and analytical procedures used should be indicated: in particular the material used for the calibration of the system.
Response 5: We added related references and a description about the materials used for the calibration to Page 4, line 151.
Point 6: References: items 16 and 18 are the same.
Response 6: This was a clerical error and has been corrected.
Point 7: The following reference should be also cited: K. Teruyama et al. "Novel chemiluminescent immunoassay to measure plasma aldosterone and plasma active renin concentrations for the diagnosis of primary aldosteronism" Journal of Human Hypertension https://doi.org/10.1038/s41371-020-00465-5
Response 7: Thank you for this providing information. This is now cited on Page9, line 279.
Having made these changes to our final manuscript, we would like to resubmit it.
Thank you once again for your consideration of our paper.
Yoshinori Ozeki
